# Nanodrug Delivery Systems for Acute Lymphoblastic Leukemia Therapy

**DOI:** 10.3390/ph18050639

**Published:** 2025-04-27

**Authors:** Aiyun Yang, Yuanfang Lu, Zuo Zhang, Jianhua Wang

**Affiliations:** 1Translational Medicine Laboratory, Beijing Key Laboratory of Child Development and Nutriomics, Capital Institute of Pediatrics, Beijing 100020, China; yangaiyuner43@163.com; 2Beijing Key Laboratory of Environmental & Viral Oncology, College of Life Science & Bioengineering, Beijing University of Technology, Beijing 100124, China; luyf1124@163.com (Y.L.); 13131216209@163.com (Z.Z.)

**Keywords:** acute lymphoblastic leukemia, drug delivery, therapy, nanovehicles

## Abstract

Acute lymphoblastic leukemia (ALL) is a malignant tumor caused by abnormal proliferation of B-line or T-line lymphocytes in the bone marrow. Traditional treatments have limitations. Because of their unique advantages, nanodrug delivery systems (NDDSs) show great potential in the treatment of ALL. In this paper, the pathological features of ALL, the limitations of current therapeutic methods, and the definition and composition of NDDSs were reviewed. Research strategies for the use of NDDSs in the treatment of ALL were discussed. In addition, challenges and future development directions of NDDSs in the treatment of ALL were also discussed, aiming to provide reference for the application of NDDSs in the diagnosis and treatment of ALL.

## 1. Introduction

Acute lymphoblastic leukemia (ALL) is a malignant tumor of the blood system, with an incidence of 20% in adults with acute leukemia and 80% in children with acute leukemia [1]. It is characterized by the abnormal proliferation and aggregation of juvenile lymphocytes in bone marrow and lymphoid tissue. The incidence of ALL peaks between the ages of 1 and 4, and then drops sharply, reaching its lowest point between the ages of 25 and 45, and rising slightly after the age of 50, with a higher incidence in men than in women [2]. At present, chemotherapy is still the first-line treatment for ALL, but traditional chemotherapy drugs have some problems, such as poor targeting and severe toxicity and side effects, which limit their clinical application. Another conventional treatment is chimeric antigen receptor T cell (CAR T cell) therapy, which has shown great promise in the treatment of childhood ALL [2]. However, there is currently only one FDA-approved CD-19-targeted CAR-T cell product (tisagenlecleucel) available for use in patients under 25 years of age [3]. Moreover, relapse remains a major obstacle to the treatment of ALL. As a new therapeutic strategy, nanodrug delivery systems (NDDSs) have been widely investigated for use in the treatment of ALL, due to their advantages of improving drug targeting, reducing toxic side effects, and enhancing therapeutic effects.

This paper aims to review the latest research strategies regarding the use of NDDSs in the treatment of ALL, discusses their application advantages, challenges, and future development directions, and provides new ideas and a theoretical basis for the precision treatment of ALL.

## 2. ALL

ALL is a common malignant disease of the blood system. It is mainly characterized by abnormal clonal proliferation of immature T and B lymphocytes, infiltrating the bone marrow, blood, or other tissues and organs, and causing abnormal hematopoietic function of the bone marrow. The clinical manifestations of ALL are diverse and involve multiple systems. The main manifestations are anemia, fever, bleeding, lymph node, liver and spleen enlargement, and bone and joint pain. ALL is the most common tumor in patients under 15 years of age, and its incidence in childhood is much higher than that of acute myeloid leukemia. ALL accounts for 20% of acute leukemia in adults and 80% of acute leukemia in children [1]. The etiology and pathogenesis of ALL are not clear, may be caused by genetic factors, environmental factors, viral infection, radiation factors, and chemical factors.

The main pathophysiological features of ALL include the following aspects: it involves extensive and uncontrolled abnormal proliferation of leukemia cells, which can infiltrate tissues and organs of the whole body, in addition to involving the hematopoietic system; some leukemia stem cells (LSCs) in ALL are in a relatively resting state, and can escape the killing of most cell cycle-specific cytotoxic drugs and lurk in the body for a long time, becoming the root of relapse; ALL has a complex bone marrow microenvironment, where ALL cells interact with the bone marrow hematopoietic microenvironment, and the proliferation of ALL cells and the progression of the disease depend on the bone marrow microenvironment; and there is a blood–bone marrow barrier (BMB) between the blood circulation and bone marrow, which is very important for allowing blood cells to enter the circulation regularly. In addition, it can selectively “trap” nutrients, drugs, and other substances, resulting in unsatisfactory clearance effect of residual LSCs in bone marrow.

Combination chemotherapy is the foundational treatment for ALL. The current chemotherapy regimen for ALL patients mainly consists of induced remission, consolidation therapy, and maintenance therapy phases. Although chemotherapy can cure 80% of ALL patients, the systemic toxic side effects related to chemotherapy are still a difficult problem affecting the survival rate and quality of life of ALL patients. Hematopoietic stem cell transplantation (HSCT) is another important treatment method for ALL patients, and is of great value in the treatment of refractory and recurrent ALL [4]. HSCT is usually decided on the basis of the patient’s condition after the combination chemotherapy has achieved a complete response. But relapse is still the leading cause of transplant failure, and analysis of data from patients with relapsed T-ALL found that patients who had not received high doses of chemotherapy had little chance of a cure after autologous or allogeneic HSCT [5]. For refractory/recurrent ALL, with rechemotherapy and retransplantation, the overall survival rate is only 10% to 20%, and the prognosis is poor. With the advent of immunotherapy, there is hope for patients with refractory/recurrent ALL. At present, the immunotherapy used to treat refractory and relapsing ALL is CAR-T, and the re-complete response rate of refractory/relapsing B-ALL can reach 90% [6,7]. However, there are still some problems with CAR-T cell therapy, such as adverse reactions encephalopathy, cytokine release syndrome (CRS), and graft-versus-host disease (GVHD) [8]. Moreover, CAR-T cell therapy currently has a rather high recurrence rate [9], so it cannot be used as a radical treatment. Therefore, more novel and effective treatment strategies for leukemia still need to be developed.

## 3. Nanodrug Delivery Systems

An NDDS refers to the use of nanotechnology to encapsulate or adsorb drugs in nanoscale vehicles to achieve targeted delivery and controlled release of drugs. The complete NDDS consists of three main parts, including the nanovehicle type, drug loading, and surface modification (Figure 1).

Nanovehicles are the skeleton of the NDDS, and are mainly divided into organic nanomaterials and inorganic nanomaterials, according to the different properties of materials (Figure 2). The most common organic nanovehicles include polymer nanoparticles, liposomes, proteins, and peptide-based nanoparticles. Commonly used inorganic nanovehicles are metals, mesoporous silicon oxide, carbon tubes, and quantum dots [10,11,12]. Compared with organic nanomaterials, inorganic nanomaterials have a variety of physicochemical properties that are closely related to their size and composition. However, inorganic nanovehicles are non-biodegradable and cannot deliver highly effective drugs, so in most cases, they must be combined with organic materials [10,13,14].

The core element of the NDDS is the loaded drug, mainly including chemotherapy, protein, peptide, genes, or the combination of two or more drugs, which can achieve better efficacy through synergistic action. In addition, diagnostic imaging agents are integrated into nanovehicles for co-delivery with chemotherapy drugs, which can be used in tumor diagnosis and cancer cell metastasis monitoring. In order to obtain better delivery results, suitable nanovehicles should be selected to encapsulate different drugs, according to the nature of the drugs and the site of action. For example, proteins and peptides generally have short half-lives in circulation, and are quickly eliminated or degraded after intravenous administration. Polymer nanoparticles are ideal drug delivery nanovehicles for protein and peptide drugs [15]. Nucleic acid drugs have poor stability in vivo and poor ability to penetrate cell membranes, due to their negative charge, high molecular weight, and strong hydrophilicity [16]. Liposomes [17], polymer nanoparticles [18], and cationic vehicle-based inorganic nanoparticles [19] are the three most common types of non-viral nanovehicles used for nucleic acid drug delivery.

Although some drug delivery systems have advantages of low autoimmune stimulation, stability, low toxicity, and low cost, their targeted delivery may remain suboptimal. Therefore, in order to achieve better therapeutic effects and reduce toxic effects of drugs, targeted molecules such as peptides, antibodies, and ligands are often modified on the surface of nanovehicles [20].

## 4. Research Strategies for the Use of NDDSs in the Treatment of ALL

### 4.1. Targeted Delivery Strategy: Enhancing Drug Specificity

#### 4.1.1. Active Targeting Modification

ALL cell surface markers (such as CD71, CD3, CD19) are specifically recognized by targeted ligands (such as antibodies, peptides, and aptamers) which are modified on the surface of nanovehicles. Krishnan et al. used a CD19-targeting ligand to modify hydrophilic poly(ethylene glycol) (PEG) and hydrophobic poly(ε-caprolactone) (PCL) polymer nanoparticles (81 nm) containing doxorubicin (in vitro experiment: 100 nM and 1 μM; in vivo experiment: 2.5 mg/kg), which could be specifically delivered to leukemia cells. CD 19 modification enabled nanoparticles to be absorbed through clathrin-dependent endocytosis, enhanced the accumulation of doxorubicin in cells, and induced apoptosis [21]. The CD19/CD3 bispecific antibody Blinatumomab has been used to treat ALL by binding T cells and leukemia cell surface antigens [22]. Durfee et al. prepared 1,2-Distearoyl-sn-glycero-3-phosphocholine (DSPC)/chol/DSPE-PEG_2000_-NH_2_ lipid bilayer structures (137 nm) supported by mesoporous silica nanoparticles. They wrapped the anticancer drug gemcitabine (0–30 μM) in it, and then modified the nanocapsules with anti-epidermal growth factor antibody as the targeting agent. The results showed that the nanocapsules had high specificity and a better therapeutic effect on leukemia cell lines compared with the nanocapsules without anti-epidermal growth factor antibody as the targeting agent [23].

ALL cells originate in the bone marrow and spread throughout the blood. Bone marrow plays an important role in the progression of ALL, so it is considered a target for ALL treatment. Developing a bone marrow targeting system could enable more drugs to be brought to the lesion site for a better antitumor response. Some abnormal signals in pathologic bone marrow have been extensively studied, such as overexpression of the chemokine CXCR4, tissue hypoxia, and increased reactive oxygen species (ROS) levels. These could be used as sensitive release conditions for the design of bone marrow-targeted drug nanovehicles. Xu et al. used polymer DSPE-mPEG2000 micelles to deliver chemically synthesized CXCR4, a chemokine receptor highly expressed in a variety of leukemia cells and strongly associated with drug resistance and relapse, to antagonize polypeptides and doxorubicin (0–20 μM). The micelles (20 nm) could effectively bind to leukemia cells expressing CXCR4, downregulating the phosphorylation of Akt-, Erk-, and Mcl-1-signaling proteins mediated by the CXCR4/CXCL12 axis, and significantly reducing leukemia cells in the peripheral blood and bone marrow, as well as spleen and liver infiltration of drug-resistant and refractory leukemia mice, and thus significantly prolonging survival [24].

LSCs, which exist in bone marrow niches, are an important factor in the relapse of leukemia resistance. Targeting these niches could activate LSCs, make them sensitive to treatment, and enhance drug clearance. It has been found that the TIM-3 protein is highly expressed on the surface of LSCs. Chen et al. synthesized a 244 nm protein nanogel loaded with the immune checkpoint blocker aCD47 and the TLR7/8 agonist Resiquimod, and modified TIM-3 targeting antibodies on its surface. The nanoparticle could actively target LSCs in the blood circulation, and achieve efficient drug enrichment in bone marrow niches, significantly inducing immunosuppression by activating the “eat me” signal of macrophages, and enhancing the anti-leukemia immune response to overcome the problem of relapse of leukemia resistance [25].

The low-hemoperfusion nature of bone (also known as the BMB) prevents the passage of drugs, resulting in unsatisfactory eradication of residual LSCs in bone marrow. Xue et al. constructed a bisphosphonate (BP) lipid nanomaterial (about 100 nm) with high affinity for bone minerals as a means of overcoming biological barriers to efficiently deliver mRNA therapeutics to the bone marrow microenvironment in vivo [26]. In addition, methods used in other biological barriers, such as modifying the penetrating TAT peptide (HIV transcription activator-derived peptide) on the surface of nanovehicles to promote transmembrane transportation, also holds promise for future applications across the BMB.

#### 4.1.2. Passive Targeting

The passive targeting strategy of NDDSs makes use of the microenvironment characteristics of tumor tissues or lesion sites (such as increased vascular permeability and loss of lymphatic drainage), so that the nanoparticles can naturally accumulate in specific areas, thereby increasing the concentration of drugs at the lesion site and reducing damage to normal tissues [27,28]. Bone marrow infiltration is an important pathological feature in ALL, and vascular permeability in the bone marrow microenvironment is significantly higher than that in normal tissues. Nanoparticles (typically 10–200 nm) passively accumulate around leukemia cells in bone marrow or lymphoid tissues through enhanced permeability and retention (EPR) effects. For example, liposomes or polymer nanocarriers (such as PLGA) enter the bone marrow through the blood circulation and accumulate around leukemia cells, extending the duration of drug action [29,30]. The sustained release function of nanocarriers can maintain an effective concentration of drugs in the bone marrow and reduce the need for frequent administration.

### 4.2. Intelligent Responsive Delivery System: Precise Control of Drug Release

Combined with the physical and chemical properties of the ALL microenvironment (such as pH, enzyme activity, and oxidative stress level), stimulus-responsive nanovehicles are designed to achieve spatiotemporal controlled release of drugs. There are two types of stimulation-responsive drug release. One involves the nanovehicles themselves being stimulation-responsive, such as pH-responsive polymer nanoparticles. When stimulated by external signals, they produce physical or chemical changes, such as swelling, solubility, dissociation, and other behaviors, so as to release drugs [31,32]. The other involves modifying stimulation-responsive groups or molecules on the internal and external surfaces of nanoparticles, and using reversible binding between the modification groups and drugs, or setting up “gated” molecules on the particles, to achieve controlled release of drugs under different types of stimulation. The stimulus signal may be exogenous, such as temperature and light, or endogenous, such as redox [33,34,35]. Premature release of drugs is effectively avoided by constructing stimulation-responsive intelligent nanodrug carriers, which significantly reduce the toxic side effects caused by drug leakage during transportation. Wei et al. prepared a folate receptor-targeted and glutathione (GSH)-responsive polymer prodrug nanoparticle (170~220 nm) by coupling 6-MP (0.1–50 μg/mL) with carboxymethyl chitosan via GSH-sensitive carbonyl vinyl sulfide. In the high-GSH environment of leukemia, the drug release in tumor cells was significantly improved, and the inhibition rate of leukemia cells was higher, while their cytotoxicity was lower in normal cells [36]. Wang et al. synthesized an amphiphilic pH-sensitive poly (ethylene glycol) methylether-B-(polylactic acid copolymer (B-aminoester)) with a size of 158 nm to support paclitaxel (PTX, 0–10 μg/mL) to improve the therapeutic effect of leukemia. The release of PTX from the micelles was significantly accelerated by decreasing the pH from 7.4 to 5.0, and the plasma circulation time was significantly prolonged. Compared with free PTX, the cytotoxic effects of PTX micelles on leukemia cells at all time points were significantly improved [37].

### 4.3. Overcoming Drug Resistance

Drug resistance of leukemia cells is one of the main causes of leukemia treatment failure [38,39]. Chemotherapeutic drugs are co-loaded with resistance reversal agents (such as P-glycoprotein inhibitors) or small-molecule inhibitors on the same nanovehicles to reduce drug efflux efficiency. For example, polymer nanoparticles can be loaded with both paclitaxel and P-gp modulator tariquidar, enhancing intracellular drug accumulation [40]. MRX-2843, a small-molecule MERTK and FLT3 kinase inhibitor currently involved in clinical trials for the treatment of relapsed/refractory leukemia and solid tumors, has been found to have a strong synergistic effect with vincristine. Kelvin et al., who developed a two-drug lipid (composed of DSPC, distearyl phosphatidyl glycerol (DSPG) and cholesterol lipids) nanoparticle (about 100 nm), found that MRX-2843 (0–800 nM) increased the sensitivity of early T cell precursor acute lymphoblastic leukemia (ETP-ALL) cells to vincristine in vivo [41]. The delivery of histone deacetylase inhibitors (HDACis) [42] or DNA methyltransferase inhibitors [43] through epigenetic regulation has also been studied to reverse epigenetic abnormalities in leukemia cells and restore chemotherapy sensitivity. Studies have shown that chemotherapy resistance and recurrence of leukemia are associated with impaired bone marrow immunosurveillance [44,45,46]. Li et al. developed an L-phenylalanine-based metabolic reprogramming immune surveillance activation nanomedical drug (MRIAN, 80 nm) to disrupt the immunosuppressive function of myeloid-derived suppressor cells in bone marrow, which effectively targeted bone marrow and activated immune surveillance in ALL. In ALL mouse models, MRIAN loaded with doxorubicin (Dox, in vitro experiment: 0.5 μM; in vivo experiment: 3 mg/kg) specifically targeted leukemia cells, but did not affect normal hematopoietic cells, thus significantly enhancing antitumor efficacy and limiting bone marrow damage and cardiotoxic side effects of doxorubicin therapy [47].

### 4.4. Multi-Mechanism Synergistic Therapy

#### 4.4.1. Synergistic Gene Therapy

Synergistic gene therapy is a promising anti-leukemia strategy. For example, nanovehicles have been used to deliver siRNA to target oncogenes (such as BCL11B and STAT5A) in order to inhibit the proliferation of ALL cells and induce apoptosis [48,49]. In recent years, with the rise of gene editing technology, the use of nanovehicles to deliver gene editing systems has attracted more and more attention. The gene editing delivery system based on DNA nanomachines developed by Tang Wantao et al. showed excellent tumor targeting and biocompatibility, and produced significant gene editing and therapeutic effects on living tumors [50]. A novel pegylated phospholipid-based cationic lipid nanoparticle (composed of ionizable lipid, DSPC, cholesterol, dimyristoyl-rac-glycero-3-methoxypolyethylene glycol (DMG-PEG), and DSPE-PEG) delivery system (71–80 nm) that compresses and encapsulates Cas9/sgRNA plasmids has been constructed for efficient and safe delivery of CRISPR/Cas9 systems in vitro and in vivo [51].

#### 4.4.2. Immunosynergistic Therapy

Immunotherapy has become a powerful clinical strategy for the treatment of leukemia. Checkpoint inhibitors are the most studied class of immunotherapy to date. Li Q et al. designed a nanoplatform (110 nm) based on glycinic acid (0–100 μM) to enhance the blocking of PD-1/PD-L1 and improve the immune response of T cells to leukemia [52]. Han X et al. designed and constructed a combined delivery platform based on platelet-hematopoietic stem cell-anti-programmed death-1 antibody (aPD1) conjugate. Due to the homing ability of hematopoietic stem cells to bone marrow, the assembly of a hematopoietic stem cell–platelet–aPD1 system effectively delivered aPD1 in a mouse model of leukemia [53]. CAR-T therapy is highly effective in relapsed/refractory ALL, but there is a risk of CRS and off-targeting. Nanoparticles optimized the delivery of CAR-T cells, such as the delivery of CAR genes via mRNA-LNP (80 nm), achieving high CAR expression and significant cytotoxicity to leukemia cells in vitro [54]. In addition, the delivery of leukemia-associated antigen WT1 and adjuvant CpG oligonucleotide via a nanodelivery system (610 nm) activated antigen presenting cells and T cell responses, enhancing the DC-mediated anti-leukemia immune response [55].

#### 4.4.3. Integrated Diagnosis and Treatment Platform

The core of the integrated diagnosis and treatment with NDDSs involves integrating the diagnosis and treatment functions into the same nanoplatform to achieve the synergistic effect of accurate disease detection and targeted therapy. For example, surface-enhanced Raman scattering (SERS)-based nanoparticles (60 nm) were used to detect early leukemia cell lesions and accurately distinguish between live, apoptotic, and necrotic leukemia cells [56]. Intraoperative real-time navigation was realized by labeling tumor cells with near-infrared fluorescence (NIR) quantum dots (10–12 nm) [57]. Magnetic iron oxide nanoparticles combined with chemotherapy drugs were used to achieve simultaneous treatment and efficacy monitoring [58]. The research strategies for the use of NDDSs in the treatment of ALL are summarized in Figure 3.

## 5. Statistics on the Application of NDDSs in ALL

Using “Acute lymphoblastic leukemia” and “Drug delivery” as key words, we searched in PUBMED, and a total of 316 articles were retrieved. The number of papers published in this field has increased significantly over the past two decades (Figure 4). However, in general, compared with other fields, there are relatively few research articles, with more attention paid by hospitals and hematology institutes. The representative applications of NDDSs used in ALL are summarized in Table 1, which shows that the nanovehicle types are mainly polymer, liposome, protein, and inorganic nanoparticles, and most studies are still mainly focused on the cellular level, with a lack of more in-depth studies at the animal and patient levels. The specific research progress is summarized below.

## 6. Clinical Translation and Challenges of NDDSs

According to reports in the literature, as of now, 16 antitumor nanomaterials have been approved for marketing, including 8 liposomes, 3 polymer micelles, and 5 nanoparticles (2 inorganic nanocarriers) [68]. Due to the variety of blood tumors and complex subtypes, there are a few nanodrugs applicable to blood tumors so far. The dual drug liposome (composed of DSPC, DSPG, and cholesterol lipids) preparation of cytarabine and daunorubicin in a 5:1 fixed drug ratio (trade name Vyxeos, also known as CPX-351) was approved by the FDA in 2017 for the treatment of clinical AML patients. The vincristine sulfate sphingomyelin liposome (Marqibo) nanodrug was approved by the FDA in 2012 for the treatment of Philadelphia chromosome-negative (Ph) adults with acute lymphoblastic leukemia (ALL) who have relapsed or progressed after two or more anti-leukemia treatments. In addition, there is a large number of nanomedicines in the clinical research stage, involving nearly 200 clinical trials [68,69,70]. Among these preclinical nanomedicines, relatively simple liposomes and micelles still dominate, and most of these drugs are still in the early stages of clinical trials, Phase I or II. Nanomedicines that have been approved clinically, and some examples of nanomedicines currently under clinical investigation for the treatment of leukemia, are shown in Table 2.

Although the clinical transformation of nanomedicines has made great progress, the biosafety of nanomedicines has attracted much attention, due to their different pharmacological characteristics, tissue distribution, and safety, resulting from the reassembly of various components of nanomedicines. The size, shape, specific surface area, surface charge, physical and chemical structure and aggregation state of nanomaterials will affect the absorption and metabolism of the drug and the drug itself, and consequently produce different levels of human cytotoxicity [71]. Although a lot of research has been conducted on the toxicity of nanoparticles, due to the wide range of types of nanoparticles, the toxicities of different types of nanomedicines are not the same, and the toxicity of the same nanomedicine to different tissues or cells and different sizes of nanomedicine to the same tissues or cells will be different. The behavior and fate of nanomedicines in vivo is one of the core issues in toxicology research. Although a large number of studies have revealed some mechanisms, many key scientific problems remain unknown or controversial, especially dynamic interactions in complex physiological environments, cumulative effects after long-term administration, degradation kinetics of degradable nanomaterials, and the secondary toxicity of metabolites. Therefore, it is necessary to establish a more systematic and standard toxicological evaluation method to study the safety of nanomedicine more comprehensively and deeply. At present, the safety evaluation methods of nanodrugs are similar to those of traditional drugs. In addition to acute toxicity evaluation, based on factors such as complement activation, hemolysis, inflammation, and oxidative stress, it is more necessary to strengthen long-term toxicity analysis. Artificial intelligence and molecular simulation are also needed to construct quantitative nanostructure–toxicity relationships (physicochemical properties) and clarify the molecular mechanisms related to toxicity [72]. In recent years, some governments around the world have attached great importance to the safety of nanomedicines and established new methods and technologies for safety evaluation of nanomedicines.

In addition, due to the complex microstructure and composition of nanomedicines, their construction process involves multi-step or complex technology, and there is a lack of controllable and reproducible nanomedicine synthesis methods, resulting in their large-scale production and quality control being restricted [73,74]. Physicochemical parameters are crucial to the biological effects of nanomedicines, so the production of industrial-grade nanomedicines requires strict control of physicochemical properties between batches, and involves higher requirements for chemistry, manufacturing, and control. Advances in microfluidic technology [75], 3D printing [76], and other technologies may help to open up possibilities for large-scale production of nanomedicines in the future.

## 7. Conclusions and Prospects

Nanodrug delivery systems provide a new strategy and hope for the treatment of ALL. By improving drug targeting, reducing toxic side effects, and enhancing therapeutic effects, NDDSs are expected to overcome the limitations of traditional therapeutic methods. However, translating nanomedicines from research to clinical practice faces significant challenges. The size, shape, and surface chemistry of nanomaterials affect their distribution in vivo, cellular absorption, and overall efficacy, so, looking ahead, the prospects for programmable nanomedicine in clinical applications are bright. Continued research and development is essential to optimize nanoparticle design and enhance targeting and delivery mechanisms. New methods for safety evaluation of nanomedicines, new technologies, and policies for the safe and effective use of nanomedicine must be established through close collaboration between researchers, clinicians, and regulatory authorities. Future research should focus on improving the biosafety, therapeutic effects, and clinical transformation of nanomedicines, as well as providing more accurate and effective treatment for ALL patients. Although there are still some challenges, with the continuous progress of nanotechnology and in-depth understanding of the pathological mechanisms underlying ALL, the application of NDDSs in clinical environments has broad prospects.

## Figures and Tables

**Figure 1 pharmaceuticals-18-00639-f001:**
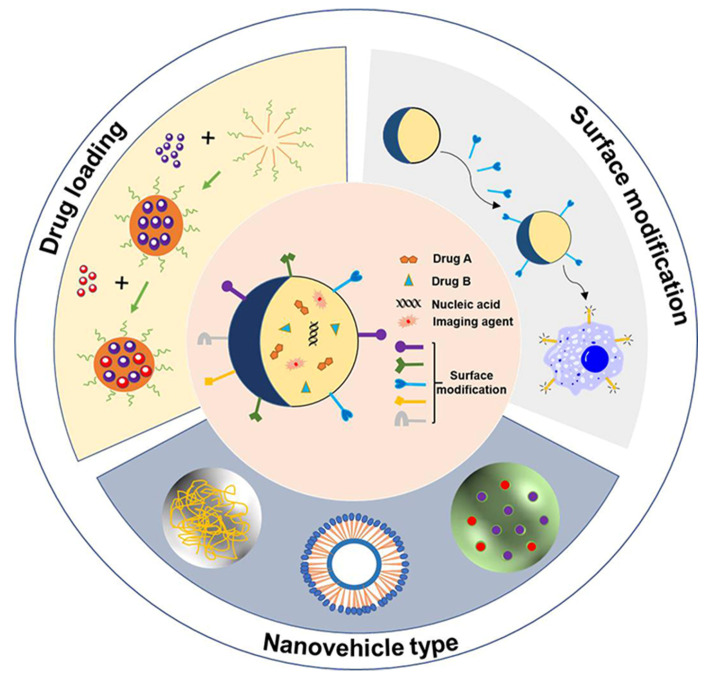
Composition of NDDS, including nanovehicle type, drug loading, and surface modification. NDDS includes various nanovehicle types that carry drugs such as chemotherapy (Drug A, Drug B), nucleic acid and imaging agents, and surface modification with peptides, antibodies, ligands, biotin and polysaccharide.

**Figure 2 pharmaceuticals-18-00639-f002:**
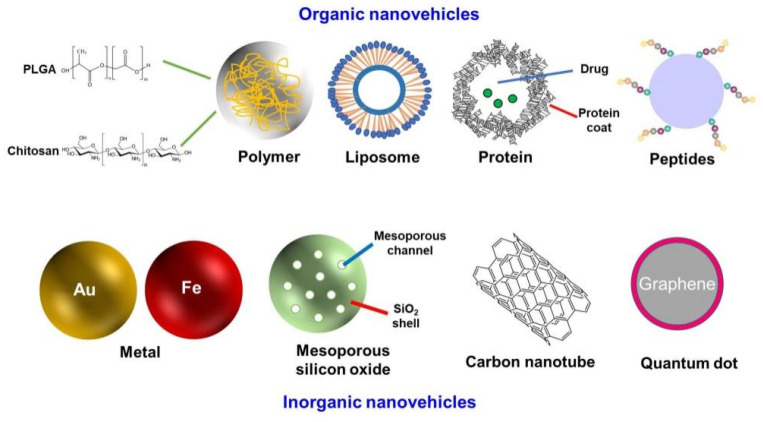
Different nanovehicles for NDDSs.

**Figure 3 pharmaceuticals-18-00639-f003:**
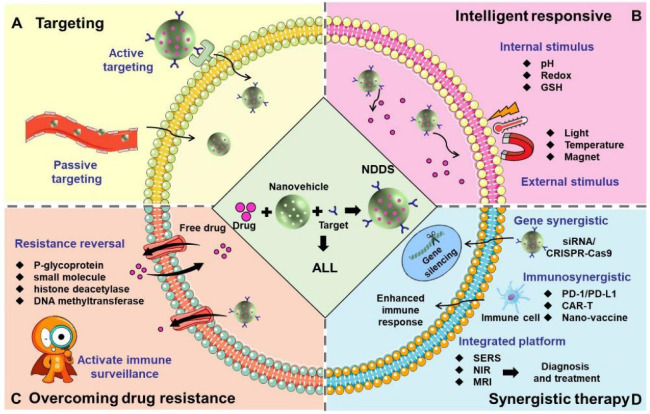
The research strategies for the use of NDDSs in the treatment of ALL. (**A**) Targeting: NDDS precisely delivers drugs to ALL cells, leukemia stem cells, the bone marrow microenvironment and blood-bone marrow barrier through active or passive targeting. (**B**) Intelligent responsive: NDDS precisely controls drug release through internal stimulus (pH, redox, GSH) or external stimulus (light, temperature and magnet). (**C**) Overcoming drug resistance: Chemotherapeutic drugs are co-loaded with resistance reversal agents (such as P-glycoprotein inhibitors), small-molecule inhibitors and histone deacetylase or DNA methyltransferase inhibitors on the same nanovehicles to reduce drug efflux efficiency. In addition, activating the immune surveillance function could also reverse drug resistance and recurrence. (**D**) Synergistic therapy: NDDS combines with gene therapy (delivering siRNA and CRISPR/Cas9 to achieve gene silencing), immunotherapy (delivering PD1/PD-L1, CAR-T, and nano-vaccines to enhance the immune response), and a comprehensive diagnosis and treatment platform (combining SERS, NIR, and MRI) to achieve multi-mechanism synergistic therapy.

**Figure 4 pharmaceuticals-18-00639-f004:**
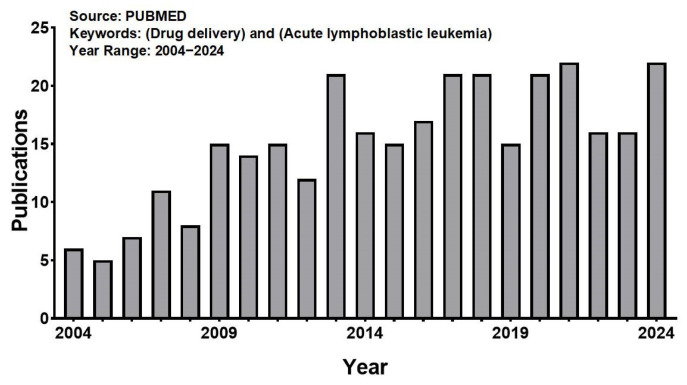
The number of publications obtained using the keywords “drug delivery” and “acute lymphoblastic leukemia” when searching the PUBMED for the period 2004–2024.

**Table 1 pharmaceuticals-18-00639-t001:** Representative applications of NDDSs used in ALL.

Vehicle	Target	Drug	Size	Application	In Vitro	In Vivo	Ref.
PEG, PCL	CD19	DOX (in vitro experiment: 100 nM, 1 μM; in vivo experiment: 2.5 mg/kg)	81 nm	Imparted cytotoxicity in ALL cells; prolonged the survival time and reduced systemic toxicity	RS4, REH	BALB/c, NSG-B2	[21]
Chitosan	-	H3TM04 (0–50 μM)	84 nm	Decreased the half maximal inhibitory concentration	Jurkat	-	[59]
mPEG-bPEI-PEBP	CD19	-	50 nm	Increased IFN-γ and IL-2; cytotoxicity also augmented	Jurkat	-	[60]
DSPG, DSPC, Chol	-	Cytarabine (0–600 mg/kg), DRB (0–9 mg/kg)	-	Superior antitumor activity and low toxicity	CCRF-CEM	Rag2-M, CB17 SCID, B6D2F1	[61]
DMPC, DOPC	-	ASNase (0–0.0081 U/L)	145–148 nm	Protected the enzyme from immune system recognition and protease degradation	MOLT-4	-	[62]
Cholesterol, phospholipids	-	mRNA (35.6 ng/μL)	70 nm	Potent cancer-killing activity	Jurkat, primary human T cells, Nalm-6	-	[63]
Casein	-	DRB (0–10 μg/mL)	127–167 nm	Prolonged DRB delivery; disrupted mitochondrial potential; inhibited cell viability	Reh	-	[64]
SiO_2_	Sgc8c	AZD5363 (0–10 mM)	69–73 nm	Targeted cancer cells more effectively; enhanced antitumor efficacy with few side effects	CCRF-CEM	-	[65]
Fe_3_O_4_	-	Genistein (0–100 μM)	12 nm	Powerful anticancer activity at a lower dose	MOLT-4,MOLT17, Jurket	-	[66]
BP-NS	Sgc8c	DOX (0–200 μg/mL)	190 nm	Targeted and synergetic chemophotothermal therapy	CCRF-CEM	-	[67]

Note: Poly(ethylene glycol) (PEG); poly(ε-caprolactone) (PCL); doxorubicin (DOX); pyrazoline (H3TM04); methoxy polyethylene glycol-branched polyethyleneimine-poly(2-ethylbutyl phospholane) (mPEG-bPEI-PEBP); distearoylphosphatidylglycerol (DSPG); distearoylphosphatidylcholine (DSPC); cholesterol (Chol); daunorubicin (DRB); 1,2-dimyristoyl-sn-glycero-3-phosphocholine (DMPC); 1,2-dioleoyl-sn-glycero-3-phosphocholine (DOPC); L-asparaginase (ASNase); black phosphorus nanosheets (BP-NS).

**Table 2 pharmaceuticals-18-00639-t002:** Nanomedicines approved or in clinical trials for leukemia.

	Name	Vehicle	Drug	Application	Year	Company	Trial Phase
Approved	Oncaspar	Polymer	L-asparaginase	ALL	1994	Enzon (Piscataway, NJ, USA)	-
	Doxil	Liposome	Doxorubicin	Multiple hematologic and solid tumors	1995	Sequus Pharmaceuticals (Menlo Park, CA, USA)	-
	Marqibo	Liposome	Vincristine	ALL	2012	Talon Therapeutics (South San Francisco, CA, USA)	-
	Vyxeos	Liposome	Cytarabine and daunorubicin	AML	2017	Jazz Pharmaceuticals (Dublin, Ireland)	-
In clinical trials	JVRS-100	Liposome	Plasmid DNA	Leukemia	2016	Milton S. Hershey Medical Center (Hershey, PA, USA)	NCT00860522 (Ph I)
	LiPlaCis	Liposome	Cisplatin, phospholipase A2	Advanced or refractory tumors	2016	Allarity Therapeutics (Wilmington, DE, USA)	NCT01861496 (Ph I)
	BP1001	Liposome	Growth factor receptor-bound protein-2 (Grb-2) antisense oligonucleotide	Leukemia	2019	Bio-Path Holdings (Bellaire, TX, USA)	NCT02923986 (Ph I) NCT02781883 (Ph II)
	Liposomal Annamycin	Liposome	Annamycin	Acute myeloid leukemia	2018	Moleculin Biotech (Houston, TX, USA)	NCT03388749 (Ph II) NCT03315039 (Ph II)

## Data Availability

No data are associated with this article.

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
