# Peer review of "Nanodrug Delivery Systems for Acute Lymphoblastic Leukemia Therapy"

_pharmaceuticals, 2025, doi:10.3390/ph18050639_

Round 1

Reviewer 1 Report

Comments and Suggestions for Authors

The review by Yang et al. discusses the development of nanostructures used as delivery systems for the treatment of acute lymphoblastic leukemia. This review is highly interesting and could be of great utility for researchers and clinical personnel working in this field. However, certain aspects require revision to make the article more comprehensive and, consequently, enhance its significance. Specifically:

  • When describing the delivery systems, the review often mentions only their nature without specifying their composition.

    • For example, in line 117, the text refers generically to "polymer nanoparticles." It would be more precise to specify the polymers used in their composition.

    • In line 122, "lipid bilayer structures" are mentioned. What is their composition?

    • In lines 183 and 207, the term "lipid nanoparticles" is used generically. The composition should be specified.

    • In lines 267 and 20, liposomes are discussed. Their composition should be specified.

    • Etc.

It is recommended to verify that, for each cited study, the composition of the nanostructures is specified and not just their nature.

  • For each type of nanostructure mentioned (e.g., nanoparticles, liposomes), it would be appropriate to include their final size in terms of dimensions.

  • In cases where drugs are encapsulated, it would also be useful to report the concentration of the drug within the nanostructures.

  • In Section 3, Nano Drug Delivery Systems, an important class of newly developed nanoparticles—peptide-based nanoparticles—is missing. The text refers exclusively to polymeric nanoparticles. This class of nano drug delivery systems should be considered and incorporated into the text. In this regard, it would be appropriate to cite the following manuscripts relevant to this topic:

    • 10.1182/blood.V126.23.3784.3784

    • 10.1038/s41598-024-60145-z

    • consider also to add a representation of peptides as organic nanovehicles in the figure 2.

  • In the Conclusions and Perspectives section, it would be advisable to discuss potential limitations associated with the use of nanostructures in this field and how these limitations could be overcome.

Reviewer 2 Report

Comments and Suggestions for Authors

The authors have compiled nano-drug delivery systems (NDDS) for acute lymphoblastic leukemia (ALL) therapy. They outlined strategies for NDDS and discussed recent trends in NDDS-based applications for ALL therapy. However, the central theme of the review article, "Applications of NDDS in ALL Therapy," is not well emphasized. The authors should provide more evidence and discuss this aspect in-detail. Additionally, incorporating more tables and figures is recommended. The reference https://doi.org/10.3390/pharmaceutics17030379 should be added in the clinical section, along with a discussion of both preclinical and clinical examples of NDDS in ALL therapy.

Reviewer 3 Report

Comments and Suggestions for Authors

Nano-Drug Delivery System for Acute Lymphoblastic Leukemia Therapy

The review article is well-written and provides a comprehensive overview of Nanoparticle Drug Delivery Systems (NDDS) for acute lymphoblastic leukemia (ALL), a topic of significant importance. The article conveys a brief overview of various nanovehicles and targeting strategies. However, following aspects need more detailed explanation.

  • The pathophysiological challenges of ALL with respect to nanoparticle systems,
  • In vivo behavior or fate of NPs under such conditions,
  • Formulation strategies to overcome these challenges,
  • Detailed mechanistic pathways involved in targeting.

Some grammatical and content-specific issues are identified.

  • Page 1, Line 29: Replace "large toxic and side effects" with "severe toxicity and side effects" for accuracy.
  • Page 4, Line 107: Rewrite the sentence: "Although some drug delivery systems have advantages due to low autoimmune, stable, low toxicity, and low cost, their targeted delivery is not ideal."

o   Instead of ‘targeted delivery is not ideal’, the authors can consider a more specific statement, ‘the targeted delivery may remain suboptimal’

o   Replace "stable" with "stability".

  • Page 5, line 183: Do you mean ‘Vincricridine’ or ‘Vincristine’?
  • Page 3, Line 101: The statement "long-acting microspheres are ideal drug delivery systems" may be misleading. Microspheres are not classified as nanosystems and are primarily utilized for controlled or sustained release, unlike nanovehicles, which are employed for cell-specific targeting. Authors should clarify this and reconsider the choice of terminology.
  • Authors refer to "modifying targeting ligands on the surface of nanovehicles." While the statement is partially correct, the correct phrasing would be "modifying the surface of nanovehicles using targeting ligands."

Overall, the topic of the review is interesting and highly relevant, and the article is comprehensive written. However, it requires major revision to address above comments.

Round 2

Reviewer 1 Report

Comments and Suggestions for Authors

The authors answered to all questions. In my opinion the revised version of the manuscript acquired redability and scientific relevance and it's now ready for the publication in the journal. Kind regards.